

# The impact of sprint interval training *versus* moderate intensity continuous training on blood pressure and cardiorespiratory health in adults: a systematic review and meta-analysis

Weibao Liang[1,*], Chuannan Liu[1,*], Xujie Yan[1], Yu Hou[2], Guan Yang[3], Jianmin Dai[4] and Songtao Wang[1]

[1] School of Physical Education and Sports Science, South China Normal University, Guangzhou, Guangdong, China
[2] Department of Physical Education, Kunsan National University, Gunsan, South Korea
[3] School of Physical Education, South China University of Technology, Guangzhou, Guangdong, China
[4] College of Sports Science, Kyungnam University, Changwon, South Korea
[*] These authors contributed equally to this work.

Corresponding author
Songtao Wang, wangsongtao@m.scnu.edu.cn

## ABSTRACT

**Background**. Although aerobic exercise is the primary modality recommended for the treatment of hypertension, it remains unclear whether high-intensity all-out sprint interval training (SIT) can result in greater reductions of blood pressure (BP) and cardiorespiratory health. This systematic review aims to compare the impact of SIT *versus* Moderate-intensity continuous training (MICT) on improvements in resting systolic blood pressure (SBP), diastolic blood pressure (DBP) and maximal oxygen uptake ($VO_2$ max) among adults.

**Methods**. We conducted a systematic search of three online databases (PubMed, Embase, and Web of Science) from January 2000 to July 2023 to identify randomized controlled trials that compared the chronic effects of SIT *versus* MICT on BP in participants with high or normal blood pressure. We extracted information on participant characteristics, exercise protocols, BP outcomes, and intervention settings. Furthermore, the changes in $VO_2$ max between the two groups were analyzed using a meta-analysis. The pooled results were presented as weighted means with 95% confidence intervals (CI).

**Results**. Out of the 1,874 studies initially were found, eight were included in this review, totaling 169 participants. A significant decrease in SBP (MD = $-2.82$ mmHg, 95% CI [$-4.53$ to $-1.10$], $p = 0.08$, $I^2 = 45\%$) was observed in the SIT group compared to before the training, but no significant decrease in DBP (MD = $-0.75$ mmHg, 95% CI [$-1.92$ to $0.42$], $p = 0.16$, $I^2 = 33\%$) was observed. In contrast, both SBP (MD = $-3.00$ mmHg, 95% CI [$-5.31$ to $-0.69$], $p = 0.68$, $I^2 = 0\%$) and DBP (MD = $-2.11$ mmHg, 95% CI [$-3.63$ to $-0.60$], $p = 0.72$, $I^2 = 0\%$) significantly decreased in the MICT group with low heterogeneity. No significant difference was found in resting SBP and DBP between SIT and MICT after the intervention. Both SIT and MICT significantly increased $VO_2$ peak, with SIT resulting in a mean difference (MD) of 1.75 mL/kg/min (95% CI [0.39–3.10], $p = 0.02$, $I^2 = 61\%$), and MICT resulting in a mean difference of
3.10 mL/kg/min (95% CI [1.03–5.18], $p = 0.007$, $I^2 = 69\%$). MICT was more effective in improving VO$_2$ peak (MD $= -1.36$ mL/kg/min, 95% CI [$-2.31$ to 0.40], $p = 0.56$, $I^2 = 0\%$). Subgroup analysis of duration and single sprint time showed that SIT was more effective in reducing SBP when the duration was ≥8 weeks or when the sprint time was <30 s.

**Conclusion**. Our meta-analysis showed that SIT is an effective intervention in reducing BP and improving cardiorespiratory fitness among adults. Consequently, SIT can be used in combination with traditional MICT to increase the variety, utility, and time efficiency of exercise prescriptions for different populations.

# INTRODUCTION

Pre-hypertension and hypertension are important risk factors for cardiovascular disease, stroke and other health problems, and hypertension affects nearly 1 billion people worldwide (*Rapsomaniki et al., 2014*; *Mills et al., 2016*; *Han et al., 2020*). Lifestyle modifications, including exercise, are recommended to prevent and treat hypertension (*Whelton et al., 2018*).

The recommended primary modality for managing hypertension is moderate-intensity continuous training (MICT) (class of recommendation I and level of evidence A) (*Rabi et al., 2020*; *Schneider, Salerno & Brook, 2020*). Meta-analyses of previous review studies have demonstrated significant reductions in the mean SBP of 6.0 to 12.3 mmHg and DBP of 3.4 to 6.1 mmHg in hypertensive individuals in response to aerobic training (*Cornelissen & Smart, 2013*; *Igarashi, Akazawa & Maeda, 2018*; *Cao et al., 2019*).

Despite recommendations from the World Health Organization (WHO) and national departments, many individuals lead inactive or sedentary lifestyles for a variety of reasons (*Sallis et al., 2016*; *Guthold et al., 2018*). The most significant barrier to physical activity is a lack of sufficient time (*Buchheit & Laursen, 2013*).

Compared to low to moderate-intensity exercise, one of the primary advantages of high-intensity interval training (HIIT) is the requirement of less exercise time while simultaneously providing similar or greater health-related benefits compared to established physical activity recommendations. HIIT is characterized by brief, high-intensity exercise that is interrupted by recovery periods. Compared with MICT, HIIT has been reported to more effectively increase aerobic capacity (VO$_2$ max) (*Tjønna et al., 2008*; *Moholdt et al., 2009*; *Ciolac et al., 2010*) and reduce risk factors associated with metabolic syndrome, including blood pressure (BP) (*Ciolac et al., 2010*), insulin action (*Tjønna et al., 2008*) and lipogenesis (*Tjønna et al., 2008*), in a variety of patient populations.

However, a large range of HIIT protocols exist that vary in duration, intensity, and volume. Given the commonly cited barrier of "lack of time" to perform exercise, considerable interest has recently been placed on extremely low-volume, time-effective

interval protocols known as sprint interval training (SIT). The program involves short, high-intensity (over 100% maximal oxygen uptake OR maximal effort) repetitions (10–30 s) alternating with recovery periods, typically 4–6 sets (*Gist et al., 2014*).

This time-efficient exercise mode has potential benefits for improving body composition (*Keating et al., 2017*), cardiorespiratory fitness (*Gist et al., 2014*; *Vollaard, Metcalfe & Williams, 2017*; *Lora-Pozo et al., 2019*), and metabolic adaptations (*Kessler, Sisson & Short, 2012*; *Jelleyman et al., 2015*). However, it is essential to distinguish all-out SIT, which has a higher proportion of anaerobic metabolism and greater neuromuscular load (*Buchheit & Laursen, 2013*), from HIIT patterns that do not reach maximal effort (*Hall, Ekkekakis & Petruzzello, 2002*).

Numerous studies have reported comparable or greater improvements in various physical health indicators with SIT, compared to conventional exercise modes such as MICT (*Gist et al., 2014*; *Weston et al., 2014*; *Keating et al., 2017*; *Way et al., 2019*). Nevertheless, the impact of SIT and MICT on blood pressure remains unclear, likely due to considerable inconsistencies in the design of SIT protocols across studies, including duration, single sprint time-to-recovery ratio, sprint rounds, and total exercise volume.

The effect of SIT *versus* MICT on VO$_2$max was assessed as the secondary outcomes of this systematic review. Cardiopulmonary fitness serves as an independent predictor of cardiovascular and all-cause mortality rates (*Lee et al., 2011*; *Ross et al., 2016*). *Lee et al. (2011)* observed that for every 3.5 milliliters/kilogram/minute (*i.e.,* one metabolic equivalent) increase in cardiopulmonary function, the risks of cardiovascular and all-cause mortality decreased by 19% and 15%, respectively. Consequently, the enhancement of cardiopulmonary fitness should be considered a primary goal (*Ross et al., 2016*).

Hence, the objective of this systematic review is to gather and evaluate all the relevant studies that examine the blood pressure and cardiorespiratory health response to SIT in normotensive and hypertensive populations and compare it to the MICT protocol.

## METHODS

The study protocol was registered in PROSPERO (CRD42023401503).

### Search strategy

The search strategy was designed by two authors (WL and SW) by an initial scoping review of the literature. Disagreement was handled by discussion with third author (XY). We conducted a systematic search of various electronic databases, including PubMed, Embase, and Web of Science, from database inception to July 2023.

The Boolean operators "OR" and "AND" were used, and the search was conducted using Mesh terms along with their respective synonyms. The Boolean search syntax displayed below was applied: "sprint interval training" OR "sprint interval exercise" OR "Sprint intermittent training" OR "sprint training" OR "sprint-interval training" OR "High-Intensity Interval Training*" OR "High intensity intermittent training" OR "High-intensity intermittent training" OR "Interval training" OR "interval exercise" OR "HIIT" OR "high intensity exercise" OR "high intensity aerobic interval training" OR "all-out exercise training" OR "all-out training" OR "all-out interval training" OR "Supramaximal

interval training'' OR ''Wingate training'' AND ''Blood Pressure*'' OR ''Post-Exercise Hypotension'' OR ''DBP'' OR ''SBP'' OR ''Hypotension'' OR ''hypertension''.

Details of the literature search strategy are available in Supplementary Material. Qualification screening of the identified studies' titles and abstracts was independently conducted by WL and CL based on the aforementioned keywords. The full text of studies that met the inclusion criteria was retrieved. Disagreements were resolved by consensus or consultation with XY.

## Study eligibility criteria

Screened studies had to meet the following inclusion criteria: (1) Randomized trials published in English; (2) analyzed human participants of both sexes, aged >18 years; (3) included SIT protocols; (4) compare the effects of SIT with moderate-intensity continuous training; (5) resting BP was the primary outcomes and $VO_2$max was secondary outcomes; (6) intervention duration $\geq$2 weeks.

Studies with interventions mixing SIT with other interventions (*e.g.*, dietary interventions, resistance training, hypoxia) and participants with a history of cardiovascular disease or other chronic conditions affecting blood pressure were excluded from the current review excluding individuals with hypertension.

Given the review's objective, it is crucial to differentiate between MICT, SIT, and HIIT.

HIIT intensity ranges between 80–100% of maximum heart rate (HRmax) or maximum oxygen consumption ($VO_2$max), while SIT intensity generally exceeds 100% HRmax/$VO_2$max (*e.g.*, 30 s of all-out exercise).The intensity of SIT typically exceeds 100% of $VO_2$max, with common exercise protocols involving 4-6 sets of 30 s of maximal effort exercise (*Gist et al., 2014*; *Naves et al., 2018*). MICT, on the other hand, generally ranges between 46–64% of $VO_2$max/HRmax or 40%–60% HR reserve/$VO_2$ reserve (*Pescatello & Medicine AC of S, 2014*).

## Data extraction

All retrieved articles were imported into Endnote X9, and duplicate articles were removed. Preliminary screening of articles was conducted by reviewing titles and abstracts, and the full texts of potentially eligible studies were downloaded and read for further assessment. Full-text studies that did not meet the inclusion criteria were excluded and reasons for their exclusion documented (Fig. 1). Data extraction from the included studies was performed using a pre-designed table, capturing the following information: first author's name, BP category, AGE, sample size, Training mode, Duration, Frequency, SIT protocol, MICT protocol. The study selection and data extraction processes were independently carried out by two reviewers (WL and CL), with mutual cross-verification. Any discrepancies were resolved through consultation with a third reviewer (XY). In cases where direct data were unavailable, authors of included studies were contacted.
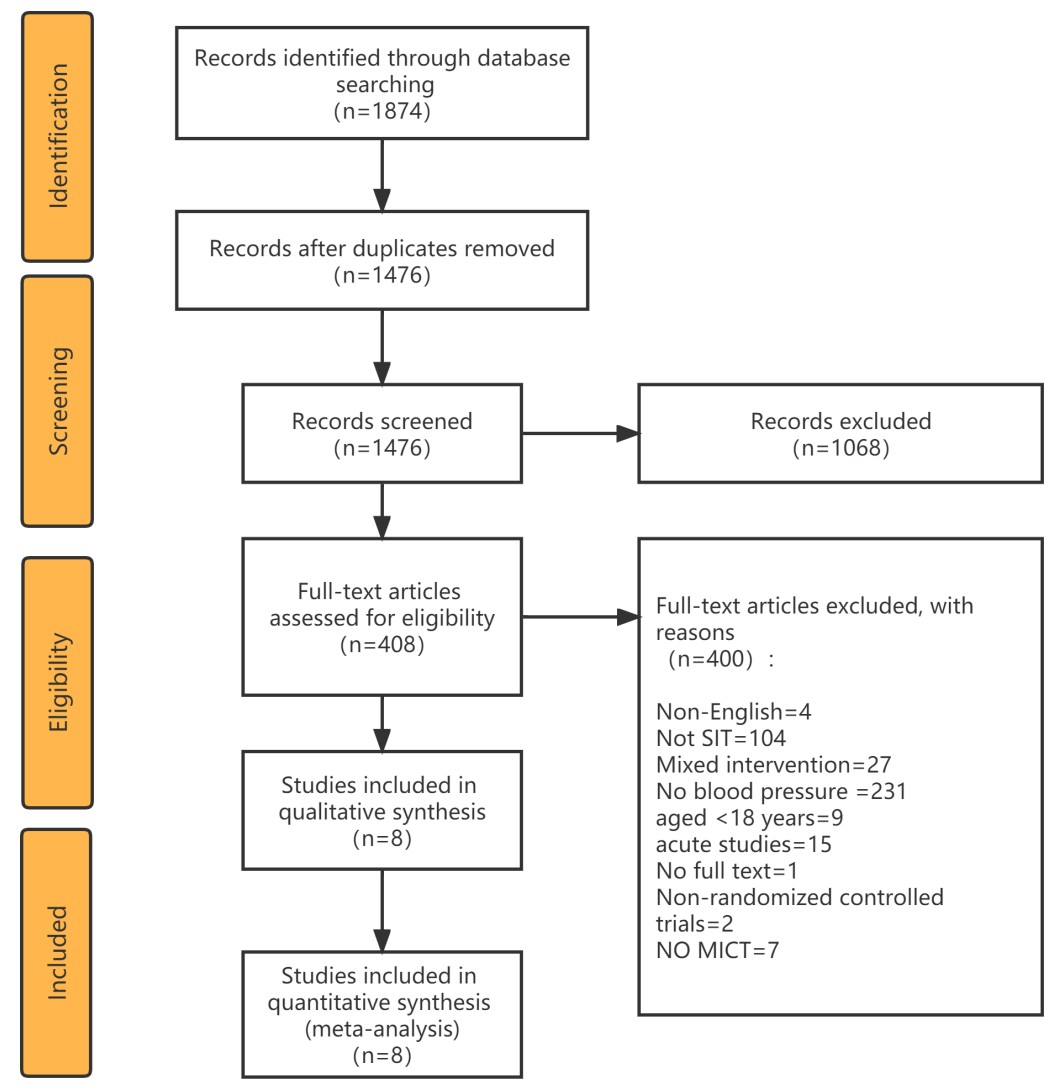

**Figure 1** Preferred Reporting Items for Systematic Reviews and Meta-Analyses (PRISMA) flowchart for study identification.

## Quality assessment and risk of bias

Two reviewers (GY and WL) independently evaluated studies that met the inclusion criteria for quality, using Review Manager 5.4 software (Cochrane Collaboration, London, UK). The tool was slightly modified to suit the study design and consisted of the following items: random sequence generation, allocation concealment, blinding of participants and personnel, blinding of outcome assessment, incomplete outcome data, selective reporting (for randomized controlled trials), and other biases. To evaluate the asymmetry of publication bias, effect sizes and standard errors were used to create funnel plots in the RevMan software.

## Statistical analyses

Statistical analyses were carried out using Review Manager 5.4 software (Informer Technologies, Inc., Los Angeles, CA, USA). Changes in the mean and standard deviation of outcome measures were used to conduct between-group and within-group meta-analyses. Change in post-intervention mean was calculated by subtracting baseline from post-intervention values. Change in the SD of post-intervention outcomes was calculated using the Review Manager. Effect sizes (ES) were measured using mean, SD, and sample size. Summary estimates with 95% confidence intervals were pooled using the DerSimonian-Laird random effects model or fixed effects model according to between-study heterogeneity (*DerSimonian & Laird, 1986*).

The $I^2$ statistic, expressed as a percentage, was used to determine heterogeneity between studies, where an $I^2 \geq 75\%$ implied high heterogeneity, $75\% > I^2 \geq 50\%$ implied moderate heterogeneity, and $I^2 < 50\%$ indicated low heterogeneity (*Higgins et al., 2003*). Forest plots were used to illustrate summary statistics and the variation across studies.

Subgroup analysis was conducted to analyze whether population and training characteristics in the study affected blood pressure. Duration and single sprint time were examined to identify any heterogeneity. Funnel plots were used to assess publication bias. Due to the limited number of included studies, univariate meta-regression analyses were not conducted.

# RESULTS

## Study selection

A PRISMA diagram of literature search and selection was presented in Fig. 1. The initial search yielded 1874 articles, 1476 of which were considered relevant after eliminating duplicates. Subsequently, 1068 studies were excluded based on assessments of their titles and abstracts. The eligibility of the remaining 408 full-text articles was scrutinized, resulting in the exclusion of 400 articles for various reasons. As a result, eight articles were included in the final meta-analysis.

## Characteristics of the studies

Table 1 presents an overview of participant characteristics. In total, 169 participants were analyzed across all studies: 84 involved in SIT and 85 in MICT. Among the included studies, three (*Rakobowchuk et al., 2008*; *Boer et al., 2014*; *Cuddy, Ramos & Dalleck, 2019*) analyzed adults of both sexes (37.5%), four (*Skleryk et al., 2013*; *Cocks et al., 2013*; *Shenouda et al., 2017*; *Petrick et al., 2021*) exclusively analyzed male participants (50.0%), and one (*Cocks et al., 2016*) exclusively analyzed female participants (12.5%).

Of the eight studies, seven (*Rakobowchuk et al., 2008*; *Skleryk et al., 2013*; *Cocks et al., 2013*; *Cocks et al., 2016*; *Shenouda et al., 2017*; *Cuddy, Ramos & Dalleck, 2019*; *Petrick et al., 2021*) included untrained or sedentary participants (87.5%), while one (*Boer et al., 2014*) did not report this information (12.5%). The participants' average age ranged from 18 to 52 years, and their BMI values ranged between 23.3 and 34.4 kg/m². Five studies (*Rakobowchuk et al., 2008*; *Cocks et al., 2013*; *Boer et al., 2014*; *Shenouda et al., 2017*; *Petrick et al., 2021*) enrolled participants with normal blood pressure, while the remaining three (*Skleryk et*

Peerj

**Table 1 Study characteristics.**

| Study | BP category | Age (y) | Sample size (% Female) | Training mode | Duration (week) | Frequence (sessions/week) SIT | MICT | SIT protocol Intensity | Recovery periods | W/R | Warm-up | Cool-down | MICT protocol |
|---|---|---|---|---|---|---|---|---|---|---|---|---|---|
| *Boer et al. (2014)* | Healthy | 18 ± 3.2 | 32 (34% F) | Cycling | 15 | 2 | 2 | 10 × 15 s 110% ventilatory threshold | 45s | 0.33 | 5 min (30W) | 5 min (30W) | 30 min at 60% ventilatory threshold |
| *Cocks et al. (2013)* | Healthy | 21 ± 0.7 | 16 (0% F) | Cycling | 6 | 3/w | 5/w | 4–6 × 30 s all-out | 4.5 min | 0.11 | / | / | 40–60 min at ∼65% VO2peak |
| *Cocks et al. (2016)* | Prehypertensive or hypertensive | 25 ± 1 | 16 (100% F) | Cycling | 4 | 3/w | 5/w | 4–7 × 30 s 200% Wmax | 4.5 min | 0.11 | 2 min (50W) | / | 40–60 min at 65% VO2peak |
| *Cuddy, Ramos & Dalleck (2019)* | Prehypertensive or hypertensive | 42.2 ± 9.7 | 32 (50% F) | Cycling | 8 | 3–5/w | 2–4/w | 2 × 20 s all-out | 3 min | 0.11 | 3 min | 3 min | 25–30 min at 40–65% HHR |
| *Petrick et al. (2021)* | Healthy | 37.4 ± 15.1 | 23 (0% F) | Cycling | 6 | 3/w | 5/w | 4–6 × 30 s ∼170% Wpeak | 2 min | 0.25 | 3 min (50W) | 2 min (50W) | 30–40 min at 60% Wpeak |
| *Rakobowchuk et al. (2008)* | Healthy | 23.3 ± 2.8 | 20 (50% F) | Cycling | 6 | 3/w | 5/w | 4–6 × 30 s all-out | 4.5 min | 0.11 | / | / | 40–60 min at 65% VO2peak |
| *Shenouda et al. (2017)* | Healthy | 27 ± 8 | 27 (0% F) | Cycling | 12 | 1–3/w | 5/w | 3 × 20 s all-out | 2 min | 0.17 | 3 min (50W) | 2 min (50W) | 45 min of cycling at 70% peak heart rate |
| *Skleryk et al. (2013)* | Prehypertensive or hypertensive | 37.8 ± 5.8 | 16 (0% F) | Cycling | 2 | 3/w | 5/w | 8–12 × 10 s all-out | 80S | 0.125 | 3 min (70W) | / | 30 min at 65% VO2peak |

**Notes.**

BP, Blood Pressure; *n*, Sample size; SIT, Sprint interval training; MICT, Moderate-intensity continuous training; w, week; W/R, Work/Rest; VO$_2$ peak, Peak oxygen uptake; Wpeak, peak power output; Wmax, maximal aerobic power; W, Watt; /, not mentioned in the paper.

*al., 2013*; *Cocks et al., 2016*; *Cuddy, Ramos & Dalleck, 2019*) included participants with pre-hypertension or hypertension. Only one (*Petrick et al., 2021*) study involved participants who were taking anti-hypertensive drugs.

Table 1 presents an overview of SIT and MICT interventions used in the included studies. MICT sessions ranged from 40 to 65% of $VO_2$peak or reserve heart rate. All four MICT studies (*Rakobowchuk et al., 2008*; *Cocks et al., 2013*; *Cocks et al., 2016*; *Shenouda et al., 2017*) lasted 40 to 60 min, while the remaining four MICT studies (*Skleryk et al., 2013*; *Boer et al., 2014*; *Cuddy, Ramos & Dalleck, 2019*; *Petrick et al., 2021*) lasted from 25 to 40 min. The SIT protocols varied in duration, frequency, and intensity across studies. All studies employed a cycling exercise that involved four (*Rakobowchuk et al., 2008*; *Cocks et al., 2013*; *Cocks et al., 2016*; *Petrick et al., 2021*) to six 30-second bouts of maximal cycling efforts, followed by a 3 to 4 min rest between bouts. Intervention durations spanned from 2 to 12 weeks, and SIT was performed three times per week in most studies, whereas MICT was performed five times per week (*Skleryk et al., 2013*; *Cocks et al., 2013*; *Cocks et al., 2016*; *Eskelinen et al., 2016*; *Shenouda et al., 2017*; *Petrick et al., 2021*) .

All studies did not mention the implementation of warm-up and cool-down protocols for MICT. Regarding SIT protocols, four studies (*Boer et al., 2014*; *Shenouda et al., 2017*; *Cuddy, Ramos & Dalleck, 2019*; *Petrick et al., 2021*) reported warm-up and cool-down durations of 2-5 min; two studies (*Skleryk et al., 2013*; *Cocks et al., 2013*) mentioned a warm-up protocol but did not specify a cool-down protocol; and two studies (*Rakobowchuk et al., 2008*; *Cocks et al., 2016*) did not provide information on either warm-up or cool-down protocols.

All studies reported blood pressure as an outcome evaluation, with Table 2 summarizing the values of SBP and DBP before and after the intervention for each study.

**Risk of bias within and across studies**

Risk of bias was evaluated for the eight studies, with most articles showing low or unclear risk in key areas. Low risk was found in a high percentage of studies for incomplete outcome data, selective reporting, random sequence, and other biases, while a moderate percentage was found in allocation concealment. However, blinding of participants and personnel, as well as outcome assessment, showed a high percentage of unclear risk across the studies. Only one study (*Boer et al., 2014*) reported blinded assessors quantifying all checked variables. Additionally, a few studies (*Rakobowchuk et al., 2008*; *Petrick et al., 2021*) showed a high risk of bias in the generation of random sequence and allocation concealment, as seen in Fig. 2.

**Meta-analysis**
**Within-group effects of blood pressure**

SIT and MICT interventions' effect on SBP and DBP levels was evaluated before and after the intervention in eight studies.

The meta analysis revealed a significant decrease in SBP (MD = −2.82 mmHg, 95% CI [−4.53 to −1.10], $p = 0.08$, $I^2 = 45\%$) in the SIT group after the intervention, with no significant changes observed in DBP (MD = −0.75 mmHg, 95% CI [−1.92 to 0.42],

**Table 2  Description of the studies included in the analysis for blood pressure effects.**

| Study | Group | Baseline | | Post-training | |
| --- | --- | --- | --- | --- | --- |
| | | SBP (mmHg) | DBP (mmHg) | SBP (mmHg) | DBP (mmHg) |
| *Boer et al. (2014)* | SIT | 124 ± 10 | 74 ± 7 | 113 ± 8 | 77 ± 8 |
| | MICT | 121 ± 11 | 72 ± 8 | 119 ± 9 | 73 ± 9 |
| *Petrick et al. (2021)* | SIT | 123 ± 13 | 78 ± 6 | 115 ± 7 | 75 ± 7 |
| | MICT | 128 ± 13 | 80 ± 9 | 119 ± 17 | 73 ± 12 |
| *Cuddy, Ramos & Dalleck (2019)* | SIT | 130 ± 9 | 83 ± 6 | 124 ± 7 | 82 ± 5 |
| | MICT | 128 ± 17 | 83 ± 10 | 127 ± 14 | 82 ± 7 |
| *Shenouda et al. (2017)* | SIT | 116 ± 8 | 68 ± 3 | 112 ± 8 | 67 ± 5 |
| | MICT | 112 ± 8 | 67 ± 5 | 111 ± 9 | 66 ± 5 |
| *Cocks et al. (2016)* | SIT | 126 ± 3 | 64 ± 2 | 125 ± 5 | 65 ± 2 |
| | MICT | 127 ± 3 | 67 ± 3 | 121 ± 5 | 65 ± 3 |
| *Cocks et al. (2013)* | SIT | 117 ± 3 | 62 ± 3 | 115 ± 3 | 59 ± 3 |
| | MICT | 114 ± 4 | 61 ± 3 | 113 ± 5 | 57 ± 3 |
| *Skleryk et al. (2013)* | SIT | 139 ± 4 | 87 ± 3 | 138 ± 4 | 84 ± 3 |
| | MICT | 142 ± 8 | 92 ± 5 | 142 ± 8 | 91 ± 6 |
| *Rakobowchuk et al. (2008)* | SIT | 112 ± 9 | 63 ± 5 | 114 ± 10 | 63 ± 6 |
| | MICT | 124 ± 14 | 66 ± 5 | 121 ± 13 | 65 ± 5 |

**Notes.**

SBP, Systolic blood pressure; DBP, Diastolic blood pressure; SIT, Sprint interval training; MICT, Moderate-intensity continuous training.

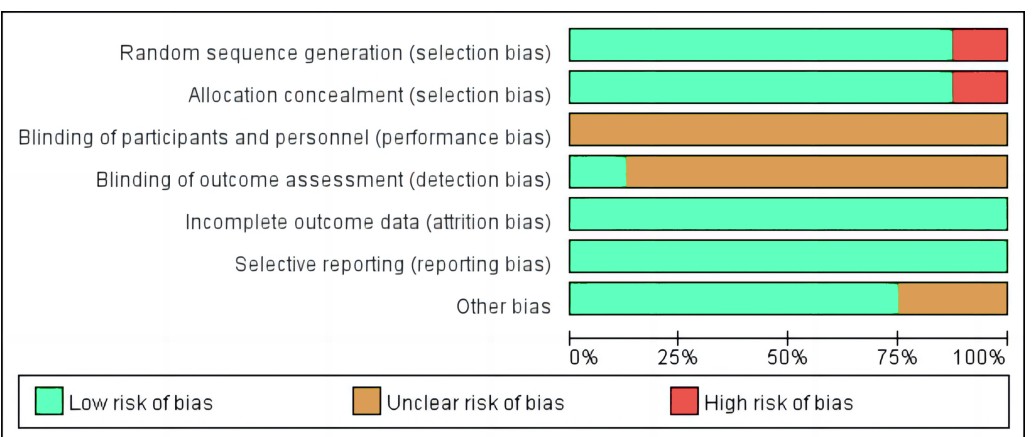

**Figure 2  Risk of bias of included studies.**

$p = 0.16$, $I^2 = 33\%$). The heterogeneity was low for both analyses ($I^2$). Forest plots of changes in resting SBP and DBP before and after intervention in the SIT group are shown in Fig. 3. Funnel plots showed no indication of publication bias (Fig. S1).

Compared to baseline values, the MICT group showed a significant decrease in both SBP (MD = −3.00 mmHg, 95% CI [−5.31 to −0.69], $p = 0.68$, $I^2 = 0\%$) and DBP (MD = −2.11 mmHg, 95% CI [−3.63 to −0.60], $p = 0.72$, $I^2 = 0\%$) levels after the intervention,

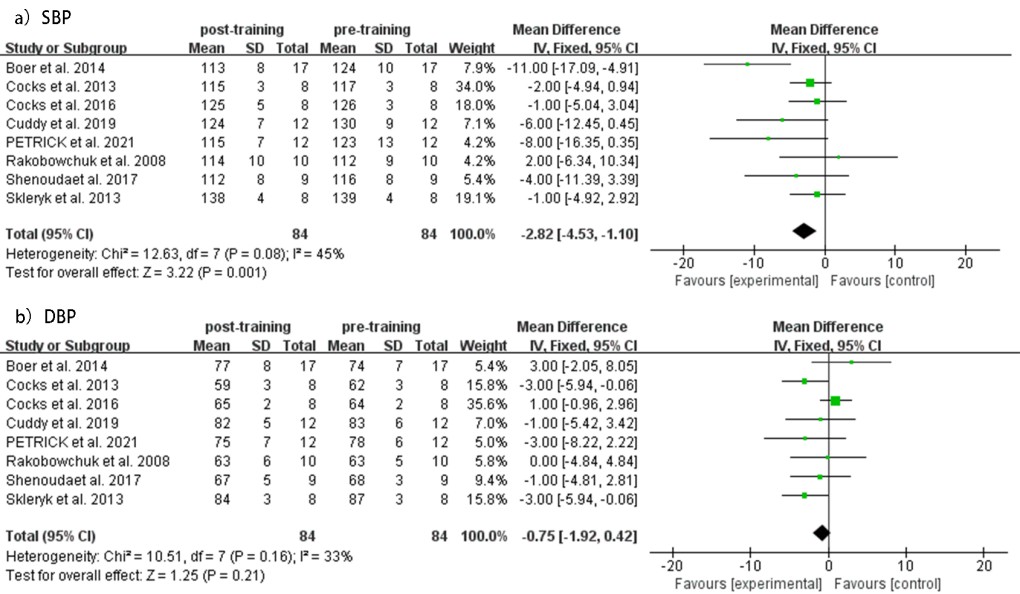

**Figure 3  Meta-analyses of the effects of SIT on BP in adults.** (A) Forest plot of eight datasets on SBP.
(B) Forest plot of eight datasets on DBP.

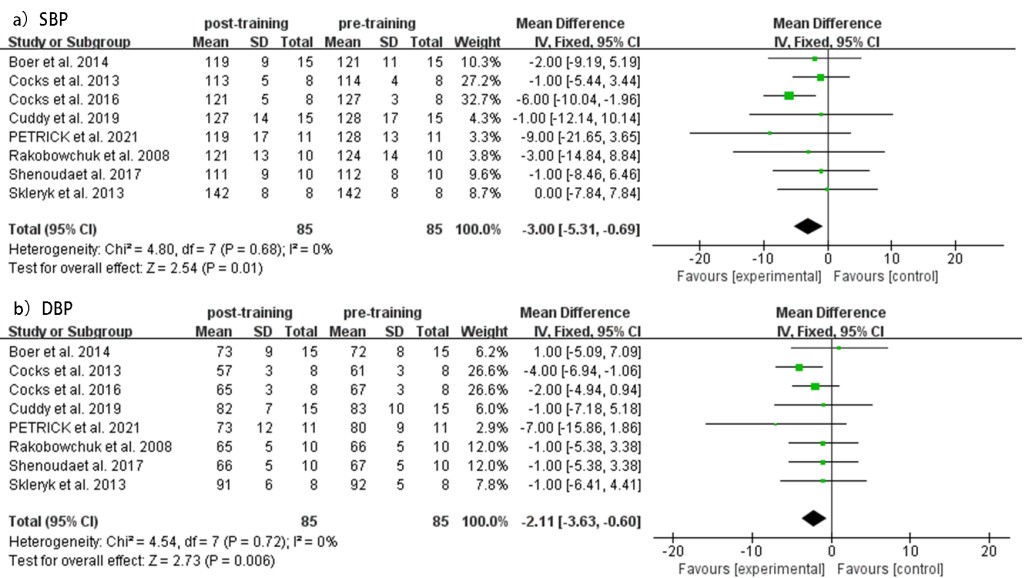

**Figure 4  Meta-analyses of the effects of MICT on BP in adults.** (A) Forest plot of eight datasets on SBP.
(B) Forest plot of eight datasets on DBP.

with low heterogeneity observed in both analyses. Forest plots of changes in resting SBP
and DBP before and after intervention in the SIT group are shown in Fig. 4. Funnel plots
showed no indication of publication bias (Fig. S2).

### Between-group effects of blood pressure

No significant differences were observed in resting SBP (MD = −0.92 mmHg, 95% CI [ −4.48 to 2.64], $p = 0.61$) changes between SIT and MICT interventions from pre- to post-intervention. However, moderate heterogeneity was detected for this analysis ($I^2 = 57\%$; $p = 0.02$). Similarly, no significant differences were found in resting DBP (MD = 1.34 mmHg, 95% CI [ −0.07 to 2.75], $p = 0.06$) between SIT and MICT interventions before and after the intervention, with lower heterogeneity observed ($I^2 = 0\%$; $p = 0.67$). The forest plots for changes in resting SBP and DBP are shown in Fig. 5. Funnel plots showed no indication of publication bias (Fig. S3).

### Cardiorespiratory fitness

SIT and MICT interventions' effect on VO$_2$peak levels was evaluated in six studies. The meta-analysis demonstrated that both SIT and MICT significantly improved VO$_2$peak (SIT, MD = 1.75mL/kg/min, 95% CI [0.39–3.10], $p = 0.02$, $I^2 = 61\%$; MICT, MD = 3.10mL/kg/min, 95% CI [1.03–5.18], $p = 0.007$, $I^2 = 69\%$). However, the pooled results of the meta-analysis suggested that MICT was more effective in enhancing VO$_2$peak than SIT (MD = −1.36 mL/kg/min, 95% CI [−2.31 to 0.40], $p = 0.56$, $I^2 = 0\%$). Forest plots for changes in VO$_2$peak are shown in Fig. 6.

### Subgroup analysis

Considering that the duration of SIT and the single sprint time may trigger blood pressure changes of different magnitudes, we performed a subgroup analysis.

### SIT *versus* MICT duration ≥8 weeks

Three studies (*Boer et al., 2014*; *Shenouda et al., 2017*; *Cuddy, Ramos & Dalleck, 2019*) investigated the effect of SIT duration greater than or equal to 8 weeks compared to MICT on SBP and DBP. The subgroup analysis of these studies indicated that when the duration was greater than or equal to 8 weeks, the SIT group showed a significant decrease in SBP (MD = −6.10 mmHg, 95% CI [ −10.51 to −1.70], $p = 0.49$, $I^2 = 0\%$) and no significant change in DBP (MD = 0.47 mmHg, 95% CI [−2.41 to 3.36], $p = 0.85$, $I^2 = 0\%$) compared with the MICT group. No heterogeneity was evident, and the forest plots of the effect of SIT duration ≥ 8 weeks *versus* MICT on SBP and DBP are shown in Fig. 7.

### SIT *versus* MICT duration <8 weeks

Five studies (*Rakobowchuk et al., 2008*; *Skleryk et al., 2013*; *Cocks et al., 2013*; *Cocks et al., 2016*; *Petrick et al., 2021*) investigated the effect of SIT duration less than 8 weeks compared to MICT on SBP and DBP. For durations less than 8 weeks, no significant changes were observed in SBP (MD = 1.71 mmHg, 95% CI [−0.71 to 4.41], $p = 0.24$, $I^2 = 28\%$), but the MICT group showed a better reduction in DBP (MD = 1.61 mmHg, 95% CI [−0.01 to 3.23], $p = 0.39$, $I^2 = 3\%$) than the SIT group. No heterogeneity was evident for this analysis. The forest plots of the effect of SIT duration <8 weeks *versus* MICT on SBP and DBP are shown in Fig. 7.

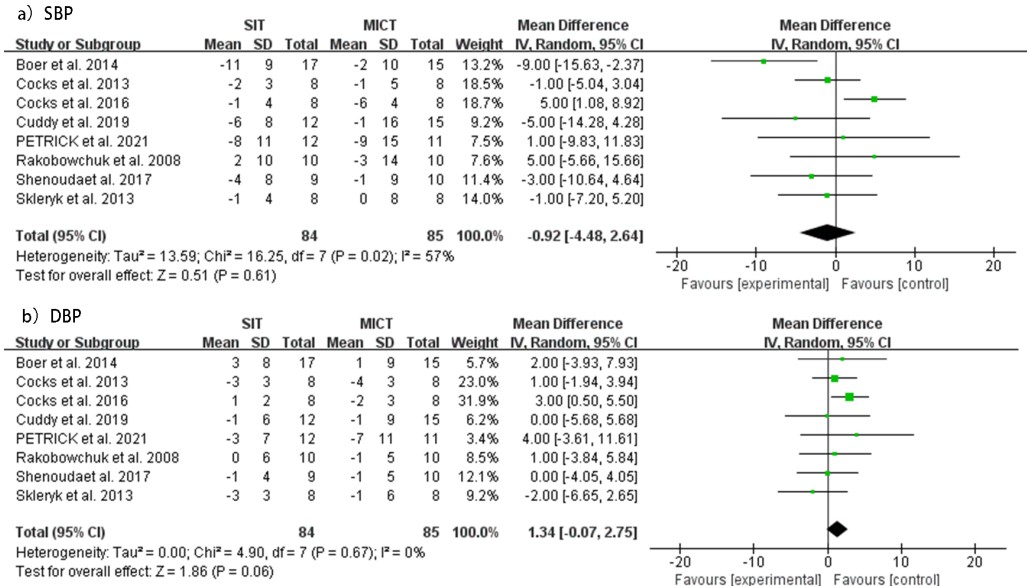

**Figure 5** **Meta-analyses of the effects of SIT *vs* MICT on BP in adults.** (A) Forest plot of eight datasets on SBP. (B) Forest plot of eight datasets on DBP.

## SIT *versus* MICT sprint time ≥30

Four studies (*Rakobowchuk et al., 2008*; *Cocks et al., 2013*; *Cocks et al., 2016*; *Petrick et al., 2021*) investigated the effect of SIT sprint time greater than or equal to 30 s compared to MICT on SBP and DBP. The subgroup analysis showed that when the sprint time was greater than or equal to 30 s, there was no significant difference between the two groups in reducing SBP (MD = 2.20 mmHg, 95% CI [−0.43 to 4.84], $p = 0.20$, $I^2 = 36\%$). However, the MICT group showed a better reduction in DBP than the SIT group (MD = 2.11 mmHg, 95% CI [0.38–3.83], $p = 0.69$, $I^2 = 0\%$). No heterogeneity was detected, and forest plots of the effect of SIT sprint time ≥30 s *versus* MICT on SBP and DBP are shown in Fig. 8.

## SIT *versus* MICT sprint time <30s

Subgroup analysis of four studies (*Skleryk et al., 2013*; *Boer et al., 2014*; *Shenouda et al., 2017*; *Cuddy, Ramos & Dalleck, 2019*) investigated the effect of SIT sprint time less than 30 s *versus* MICT on SBP and DBP. For sprint times less than 30 s, SIT was significantly more effective in reducing SBP compared to MICT (MD = −4.39 mmHg, 95% CI [−7.98 to −0.80], $p = 0.37$, $I^2 = 5\%$), while the effect on reducing DBP was similar between the two groups (MD = −0.21 mmHg, 95% CI [−2.67 to 2.24], $p = 0.77$, $I^2 = 0\%$). No heterogeneity was detected, and forest plots of the effect of SIT sprint time <30 s *versus* MICT on SBP and DBP are shown in Fig. 8.

The subgroup analysis showed that the reduction in BP in the SIT group was significantly impacted by the duration or sprint time of SIT relative to the MICT group (Table 3). For durations greater than or equal to 8 weeks, SIT was more effective in lowering SBP, whereas for durations less than 8 weeks, MICT was more effective in lowering DBP. When the sprint

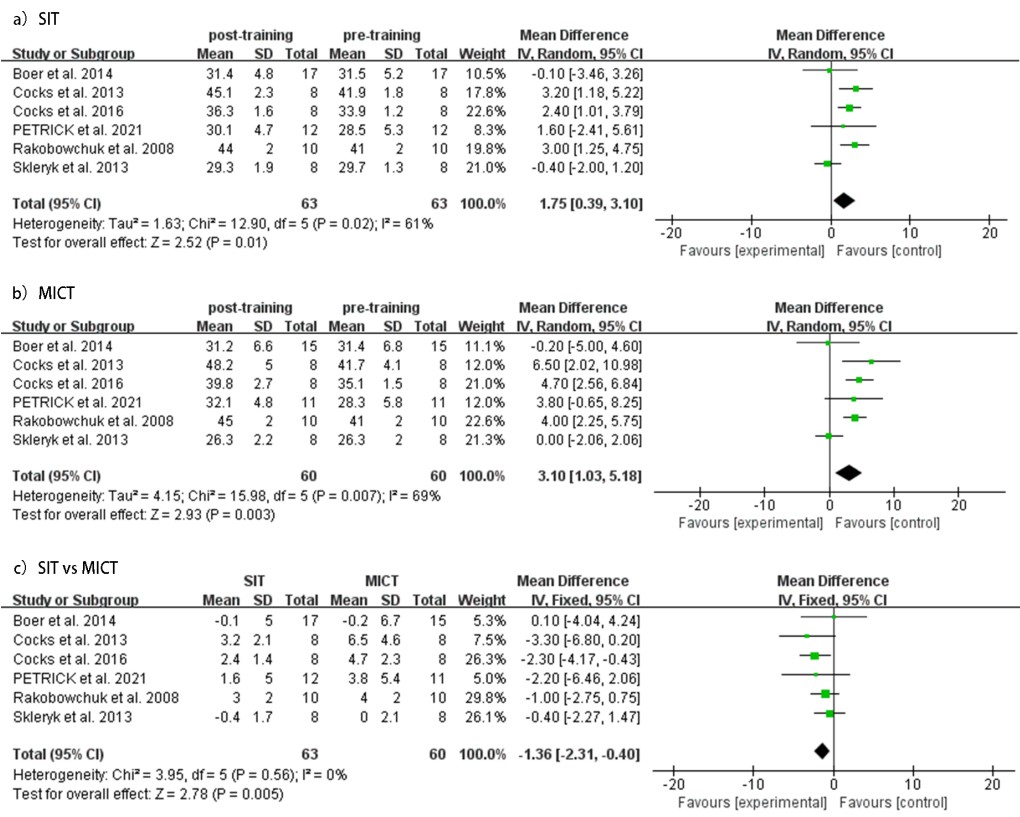

**Figure 6** **Meta-analyses of the effects of SIT and MICT on VO₂ peak in adults.** (A) Forest plot of eight datasets on VO₂ peak for SIT. (B) Forest plot of eight datasets on VO₂ peak for MICT. (C) Forest plot of eight post-intervention VO₂ peak datasets on SIT compared to MICT.

time was greater than or equal to 30 s, MICT was more effective in lowering DBP, whereas SIT was more effective in lowering SBP when the sprint time was less than 30 s.

# DISCUSSION

SIT is often promoted as an effective and time-efficient approach for improving cardiometabolic health (*e.g.*, VO₂max) (*Gist et al., 2014*) and overall health (*e.g.*, blood pressure) (*Gibala et al., 2012*) within the context of exercise training for the general population (*Gibala et al., 2006*).

This study is the first systematic review to compare the efficacy of SIT and MICT in reducing BP in adults. The main findings were as follows: (1) Exercise interventions induced similar reductions in resting SBP and DBP for both SIT and MICT. However, the limited number of included studies made it infeasible to compare the effects of SIT and MICT on ambulatory blood pressure levels. (2) MICT was more effective than SIT in improving VO₂max. (3) The subgroup analysis based on the duration and sprint time revealed that SIT was more effective in lowering SBP when the duration was ≥8 weeks, or the sprint time was <30 s. On the other hand, MICT showed a greater effect on lowering SBP when the duration was <8 weeks or the sprint time was ≥30 s.

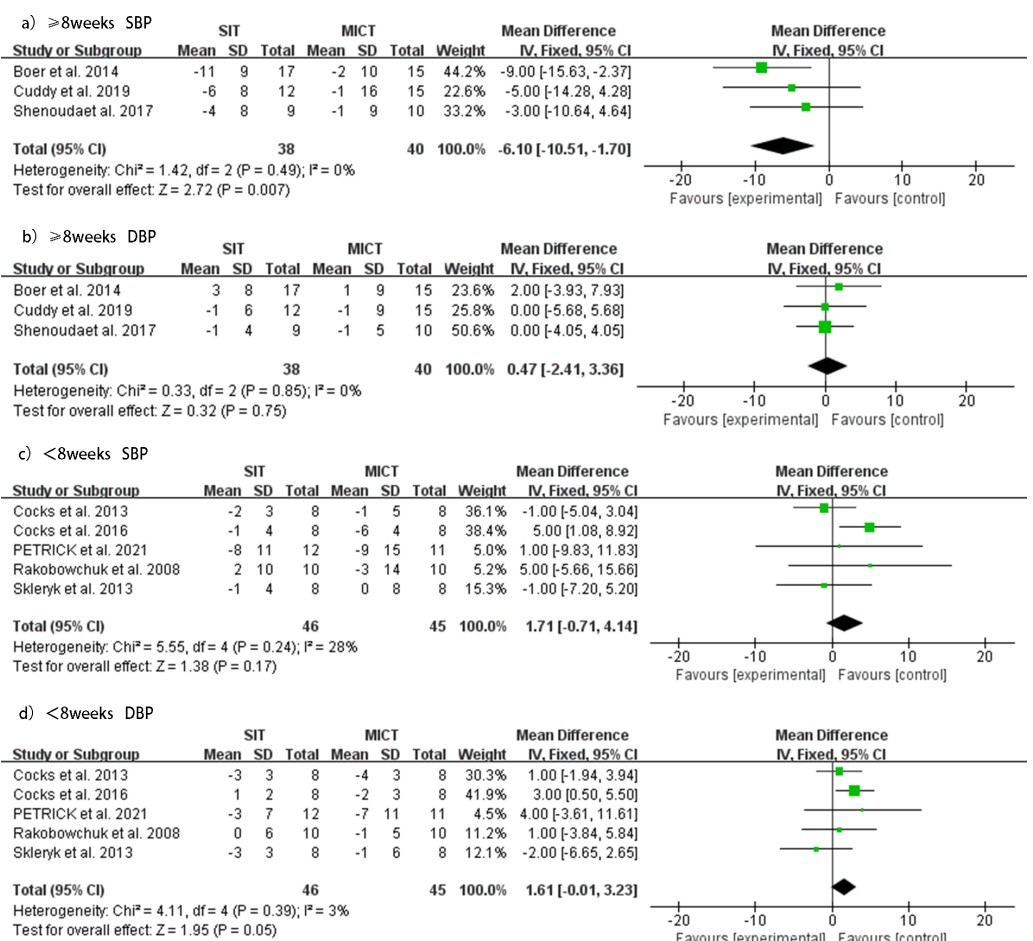

**Figure 7** **Subgroup analysis of the effects of SIT _versus_ MICT on BP in adults.** (A) Forest plot of ≥8 weeks duration study datasets on SBP. (B) Forest plot of ≥8 weeks duration datasets on DBP. (C) Forest plot of <8 weeks duration datasets on SBP. (D) Forest plot of <8 weeks duration datasets on DBP.

Resting SBP reduction was 2.82 mmHg and 3.00 mmHg for SIT and MICT, respectively, while resting DBP reduction was 0.75 mmHg and 2.11 mmHg for SIT and MICT, respectively. No significant differences in BP reduction between SIT and MICT were observed. These findings suggest that SIT and MICT produce similar reductions in BP. Comparable results were demonstrated in a previous meta-analysis (_Cornelissen & Smart, 2013_) where moderate and high-intensity aerobic exercise protocols led to reductions in resting SBP and DBP.

Cardiorespiratory fitness independently predicts both cardiovascular and all-cause mortality (_Kodama et al., 2009_; _Fardman et al., 2021_). Findings from the study showed that a 1-metabolic equivalent increase in $VO_2$max corresponded with a 13% and 15% reduction in the risk of all-cause mortality and cardiovascular disease, respectively (_Glass & Dwyer, 2007_). Therefore, clinical practitioners should consider improving cardiorespiratory fitness as a goal, particularly for individuals who are unfit.

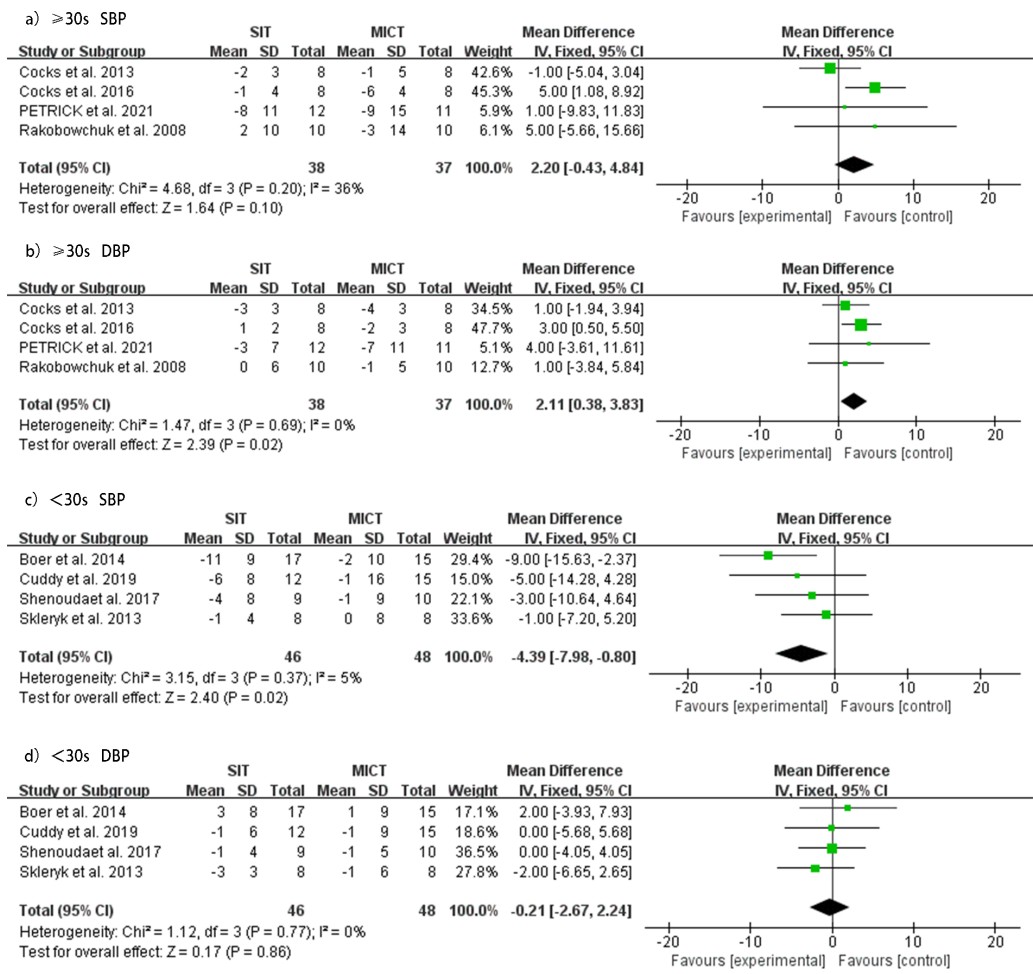

**Figure 8** **Subgroup analysis of the effects of SIT *versus* MICT on BP in adults.** (A) Forest plot of ≥30s study datasets on SBP. (B) Forest plot of ≥30s datasets on DBP. (C) Forest plot of <30s datasets on SBP. (D) Forest plot of <30s datasets on DBP.

The recommended amount of aerobic exercise for maintaining good health is generally at least 150 min of low to moderate intensity exercise or 75 min of high-intensity exercise per week (*Zhang et al., 2017*). However, *Scribbans et al. (2016)* observed that performing SIT for just 23 min for three days each week led to improvement in cardiorespiratory fitness, with effects comparable to those seen in studies using continuous exercise programs.

The secondary outcome of this systematic review was the impact of SIT and MICT on VO$_2$max. Our findings indicate that both SIT and MICT can significantly increase VO$_2$max, with MICT showing greater relative improvement.

However, the advantages of SIT for cardiorespiratory health remain apparent, consistent with prior literature (*MacInnis & Gibala, 2017*; *Petrick et al., 2021*). SIT-induced enhancements in aerobic performance have commonly been linked to muscular adaptations, such as improved muscle oxidative capacity (*Burgomaster et al., 2005*), better

functioning of muscle microvascular structure (*Cocks et al., 2013*), and a progressive shift toward type IIA muscle fibers (*De Smet et al., 2016*).

Although the studies included in this analysis designed the efficacy of MICT and SIT according to general guidelines, the training protocols differed in parameters such as rest interval, single sprint time, volume, and frequency. Therefore, we were unable to distinguish the direct effects of these protocols, which warrants further research.

A notable difference among all included study SIT was the range of interval protocols for SIT protocols. In previous literature, the range of SIT intensities and single sprint times varied, from 170% Wpeak—full intensity to 10–30 s of sprint time, and the recovery period between intervals also varied from 45 s to 4 min and 30 s.

The recovery interval may be crucial for chronic adaptation (*Cochran et al., 2014*) and metabolic response to acute episodes (*Hazell et al., 2014*). It has been demonstrated that oxygen consumption peaks during the initial 20-second recovery period after SIT and then decreases rapidly until 2 min into the recovery period (*Hazell et al., 2014*). However, the optimal recovery interval for performing SIT has not been identified. In addition to the recovery interval, the exercise volume of MICT is approximately four times higher than that of SIT, which may affect cardiometabolic outcomes. Therefore, this may limit exposure to MICT compared to optimal exercise volume and frequency guidelines.

In summary, there is considerable variability in SIT protocols. Our study, along with previous research that compared differing SIT frequencies (*Gurd et al., 2016*), suggests that frequency is a crucial factor worth exploring. Previous studies indicate that adhering to general guidelines (such as high-frequency MICT and low-frequency SIT) while considering all parameters can lead to significant benefits, such as improved blood pressure control, better systemic metabolism, and decreased risk of cardiometabolic disease. The majority of studies included in our systematic review reported an exercise frequency of five times per week for MICT and three times per week for SIT, which is consistent with the guidelines utilized in those studies. Our meta-analysis showed that MICT and SIT were almost equally effective in enhancing SBP, DBP, and VO$_2$max. However, MICT may be slightly more effective than SIT in improving diastolic blood pressure and VO$_2$max, although this might be confounded by the differences in exercise frequency.

Although using a single protocol with the same frequency provides a more direct comparison of the effects of frequency, our objective was to assess the practicality of these protocols. In reality, it is not feasible for individuals to exercise at high-intensity levels five times per week, mostly due to time constraints and limited recovery capacity among the current population.

Conversely, SIT offers the advantages of brevity and efficiency in its effects, and its potential importance in particular populations should not be underestimated (*Petrick et al., 2021*). Therefore, we recommend a hybrid MICT-SIT protocol as a means of optimizing exercise frequency and improving health outcomes.

We were unable to conduct a meta-analysis due to the limited number of studies ($n = 2$) that compared SIT and MICT on ambulatory blood pressure as an outcome. Nonetheless, it is worth noting that ambulatory blood pressure monitoring is a more accurate reflection of cardiovascular events and health than office blood pressure measurement

Liang et al. (2024), *PeerJ*, DOI 10.7717/peerj.17064

**Table 3  Description of the meta-analysis subgroups.**

| Sub-analysis | N (SIT/MICT) | Systolic blood pressure | | | | Diastolic blood pressure | | | |
|---|---|---|---|---|---|---|---|---|---|
| | | Mean difference (CI 95%) | P value | Homogeneity | | Mean difference (CI 95%) | P value | Homogeneity | |
| | | | | $I^2$ | Pvalue | | | $I^2$ | Pvalue |
| Duration | | | | | | | | | |
| ≥8 weeks | 38/40 | −5.55 [−10.51, −1.70] | 0.007 | 0% | 0.49 | 0.47 [−2.41, 3.36] | 0.0001 | 0% | 0.85 |
| <8 weeks | 46/15 | 1.71 [−0.71, 4.41] | 0.17 | 28% | 0.24 | 2.11[−0.01, 3.23] | 0.05 | 3% | 0.39 |
| Single sprint time | | | | | | | | | |
| ≥30 s | 38/37 | 2.20 [−0.43, 4.84] | 0.1 | 36% | 0.2 | −0.81 [0.38, 0.83] | 0.02 | 0% | 0.69 |
| <30 s | 46/48 | −4.39 [−7.98, −0.80] | 0.02 | 5% | 0.37 | −0.21 [−2.67, 2.24] | 0.86 | 0% | 0.77 |

**Notes.**

N, sample size; SIT, Sprint interval training; MICT, Moderate intensity continuous training.

(*Bliziotis, Destounis & Stergiou, 2012*; *Piper et al., 2015*). For this reason, we recommend conducting future randomized controlled trials (RCTs) to compare the effectiveness of SIT and MICT in reducing ambulatory blood pressure in adults.

Although our meta-analysis demonstrated the effectiveness of SIT in reducing blood pressure, the optimal SIT protocol for reducing blood pressure remains unclear. Our study's limitations include the small sample size of studies and their participants, heterogeneity in the SIT protocols and participants' characteristics, and the possibility of bias in some included studies. This review incorporates studies that were exclusively conducted through cycling. The generalizability of these findings to other forms of MICT or SIT, such as running or swimming, remains to be further validated.

Furthermore, some studies did not take into consideration the potential interference of warm-up and cool-down protocols on the overall exercise regimen when designing the warm-up and cool-down protocols for SIT. On the other hand, MICT, due to its relatively lower intensity, either did not incorporate warm-up and cool-down protocols or did not mention these protocols. Subsequent research should adhere to a consistent standard as a guiding principle in designing exercise protocols and should consider incorporating warm-up and cool-down protocols in MICT.

Therefore, our findings should be interpreted with caution. We recommend that future well-controlled RCTs be conducted to confirm our findings and determine the optimal SIT protocol for reducing blood pressure in different populations.

## CONCLUSIONS

In summary, our meta-analysis indicates that sprint interval training (SIT) is an efficient intervention for reducing blood pressure among adults. SIT presents itself as a practical and achievable alternative to conventional moderate-intensity continuous training (MICT) while promoting additional health benefits such as increased cardiovascular fitness and improved metabolic functioning. It is noteworthy that all studies included in this review implemented exercise interventions through cycling. By considering the findings of this analysis, clinicians and researchers should recognize SIT as a plausible modality for preventing and treating hypertension in adults.

## ACKNOWLEDGEMENTS

We would like to thank the researchers for their contributions.

### Funding
The authors received no funding for this work.

### Competing Interests
The authors declare there are no competing interests.

## Author Contributions

- Weibao Liang conceived and designed the experiments, performed the experiments, analyzed the data, prepared figures and/or tables, and approved the final draft.
- Chuannan Liu performed the experiments, prepared figures and/or tables, and approved the final draft.
- Xujie Yan performed the experiments, authored or reviewed drafts of the article, and approved the final draft.
- Yu Hou analyzed the data, prepared figures and/or tables, and approved the final draft.
- Guan Yang analyzed the data, authored or reviewed drafts of the article, and approved the final draft.
- Jianmin Dai analyzed the data, prepared figures and/or tables, and approved the final draft.
- Songtao Wang conceived and designed the experiments, authored or reviewed drafts of the article, and approved the final draft.

## Data Availability

The funnel plots are available in the Supplementary File.

## Supplemental Information

Supplemental information for this article can be found online at http://dx.doi.org/10.7717/peerj.17064#supplemental-information.

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
