# Peer review of "The impact of sprint interval training versus moderate intensity continuous training on blood pressure and cardiorespiratory health in adults: a systematic review and meta-analysis"

_PeerJ, doi:10.7717/peerj.17064_

## Round 0.1 · original submission · Major Revisions

Please respond to the reviewer´s comments

Regards

Dr. Manuel Jiménez

Reviewer 1 ·

Basic reporting

The SIT protocol should be clearly defined, as outlined in Gist, N. H., Fedewa, M. V., Dishman, R. K., & Cureton, K. J. (2014). "Sprint Interval Training effects on aerobic capacity: a systematic review and meta-analysis." Sports Medicine, 44(2), 269–279. https://doi.org/10.1007/s40279-013-0115-0 that it involves 4-6 repetitions of 30 seconds at maximum intensity.

Experimental design

I consider it important to define more clearly the differences when creating different SIT or MICT protocols. While it is indicated that in most studies the intensity of these protocols based on VO2peak would be important, authors should specify whether the same protocol was used (as well as whether a verification of the test results was conducted) to assess this VO2peak. It seems strange that in one study there is mention of 170% of Wpeak, and in another, Cocks et al. 2016 (Cocks et al., 2016), there is reference to 200% of Wmax. The latter (Wmax) is not defined either in the text or in Table 1 (Study characteristics), which would be necessary.
As mentioned earlier, studies with less than 4 repetitions should not be included in the analysis, and there are two studies with 2 or 3 repetitions. In this regard, the study by Cuddy et al., 2019, represents a REHIT (2 repetitions) rather than a SIT. These studies should be replaced with others where the volume is either 4-6 repetitions (SIT) or 2-3 repetitions (REHIT), avoiding the mixing of both protocols.

Validity of the findings

The explanation of the training protocols is incomplete. The warm-up and cool-down protocols for any of the studies are not provided. If the time or duration of SIT is added to its warm-up (always necessary) and potential cool-down, how much does it differ from MICT? Regarding the results, what portion of this work is at low intensity?
In the discussion and even in the conclusions, it should be made clear that both SIT and MICT were conducted through cycling, and the findings cannot be generalized to other modes of MICT or SIT.

·

Basic reporting

Methodologically coherent manuscript. Clear objective and rationale. Use scientific writing based on relevant and updated bibliography. The tables and graphs are adequate and can be read clearly, representing the results that the authors intend to express.

Experimental design

The study design is clear, but I have some observations that I detail below:

1. I suggest citing the following idea "The search strategy was designed by Weibao Liang and Songtao Wang by an initial scoping review of the literature" and using a more appropriate format to cite this idea. For example: "The search strategy was designed by Liang and Wang (year)..."

2. the following sentence is a bit ambiguous: "The Boolean operators "OR" and "AND" were used, and the search was
conducted using Mesh terms along with their respective synonyms". I suggest describing that the terms related to high-intensity exercise and blood pressure were connected in the search with the Boolean term "AND". Since previously the authors were clear and explicit in showing the use of the term "OR".

3. In the data extraction section, I suggest being more explicit in the steps taken to review the articles. They could point out at least the steps of reviewing titles and abstracts and reviewing the full-text article.

4. Why did you not evaluate the methodological quality of the articles? I think it is an important point that needs to be improved. I suggest adding it. Currently, most systematic reviews with meta-analysis evaluate the methodological quality of the articles found.

5. Why do you use ROB and not ROB2 to evaluate the risk of bias? ROB2 is currently used. Is it possible to update this evaluation?

Validity of the findings

The conclusions are consistent with the design and analysis carried out. The results are clear and the discussion is pertinent.

Additional comments

No comment

·

Basic reporting

Major Revision
The introduction is unclear and lacks a clear argument about the benefits of high-intensity exercise compared to continuous exercise. It is already known that continuous exercise affects blood pressure, but the proposal to compare it with high-intensity exercise is not explained.

Experimental design

Revise the first paragraph to enhance its clarity, as there are inconsistencies in the writing between (lines 57 and 58).

The authors do not establish a clear link between the paragraphs in line 67. It is suggested to connect the discussion of VO2 max to the previous paragraph to ensure a logical flow of information.

In line 105, it is unclear what the authors meant by 'the search strategy was devised' and 'the name of the authors'. It is important to note that the search strategy was developed by two authors and that, in case of doubts regarding the inclusion of articles, a third author was consulted.

Did the authors only use keywords related to interval or high-intensity exercise in line 108 of the search strategies (MESH)? Did they not search for studies with continuous moderate-intensity exercise?

I suggest removing the word 'relevance' from line 118 and leaving only the defined eligibility criteria.

Validity of the findings

The study's findings have implications for the literature and are analysed in accordance with the specificities of a systematic review. The authors should clarify how they determined the relevance of the article by using only the inclusion criteria proposed in the methods.

Additional comments

The article is well-designed and the results are interesting. However, adjustments need to be made as proposed.

---

## Round 0.2 · Major Revisions

Dear Author:

Please attend to all comments from reviewers.

Thank you

Reviewer 1 ·

Basic reporting

There are some errors in the wording (spacing usage), such as "W peak" (Wpeak) or "Cronic adaptation(Cochran et al., 2014)". It is recommended to review the entire text and correct any potential errors.
In the references section, there are several errors (author names, improper use of capitalization, etc.).

Experimental design

No comment

Validity of the findings

No comment

Additional comments

Thank you for the improvements made to the article. However:
-There are some errors in the wording (spacing usage), such as "W peak" (Wpeak) or "Cronic adaptation(Cochran et al., 2014)". It is recommended to review the entire text and correct any potential errors.
-In the references section, there are several errors. Review all citations and correct any inaccuracies (author names, improper use of capitalization, etc.).

·

Basic reporting

The article is relevant to the literature and all the necessary changes have been made

Experimental design

The experimental design was adequate after the reviewers' considerations.

Validity of the findings

The conclusions are interesting and are in line with the reviewers' considerations.

---

## Round 0.3 · accepted · Accept

I am writing to inform you that your manuscript "The impact of sprint interval training versus moderate intensity continuous training on blood pressure and cardiorespiratory health in adults: a systematic review and meta-analysis" has been Accepted for publication. Congratulations!

·

Basic reporting

No comment.

Experimental design

No comment.

Validity of the findings

No comment.

Additional comments

I would have hoped that the response letter to the reviewers would specifically include comments on my observations. However, I could read in the manuscript that there are improvements in the items I mentioned.